# Fine Particulate Matter Reduction through Crop Surface Adsorption in an Agricultural Field Using the Coefficient Derived from Wind Tunnel Experiments

Seong-Won Lee, Kibwika Anthony Kintu and Il-Hwan Seo *

Department of Rural Construction Engineering, College of Agriculture & Life Sciences, Jeonbuk National University, Jeonju-si 54896, Republic of Korea; afe18@jbnu.ac.kr (S.-W.L.); anthony@jbnu.ac.kr (K.A.K.)
* Correspondence: ihseo@jbnu.ac.kr; Tel.: +82-010-3474-5001

**Abstract:** Fine dust can have serious effects on human health and crop growth. Fodder crops can reduce airborne dust by coagulating soil particles and reducing wind speed on the surface and have the effect of reducing fine dust by adsorbing it on the crop surface. In this study, the dust reduction coefficient of crops was derived through a self-manufactured wind tunnel experiment to quantitatively assess the dust reduction effect of crops by type and planting density. Additionally, a dust reduction formula considering crop growth and weather conditions during the cultivation period was derived. The dust reduction coefficient was measured by the gravimetric method and the real-time size distribution of dust concentration before and after the crop. The PM reduction coefficient showed triticale at PM-2.5 82.2 mg/m$^3$, PM-10 120 mg/m$^3$, and barley at PM-2.5 14.5 mg/m$^3$, PM-10 26.9 mg/m$^3$ under moderate planting density. During the general planting density cultivation period of triticale and barley, PM-10 was reduced by 37.8 kg/ha and 8.5 kg/ha, respectively, and PM-2.5 was reduced by 25.9 kg/ha and 4.6 kg/ha. The dust reduction effect during the cultivation period was up to 126.1 kg/ha in terms of PM-10 when triticale was cultivated with densely sowing planting density.

**Keywords:** barley; fine dust; PM reduction; triticale

## 1. Introduction

The amount of fine dust emitted annually in Korea is 392,351 tons of TSP, 146,733 tons of PM-10, and 58,557 tons of PM-2.5 (Ministry of Environment, 2022). Particulate matter (PM) can have serious effects on health, such as respiratory diseases, cardiovascular diseases, and immune suppression. Many studies present the harmful effects on the human body when exposed to high concentrations of PM for short periods or to low concentrations over long periods [1–4]. Both PM-2.5 and SO$_2$ were positively correlated with all-cause, lung cancer, and cardiopulmonary mortality, and every 10 μg/cm$^3$ increase in PM-2.5 was associated with a 4, 6, and 8% increased risk of all-cause, cardiopulmonary, and lung cancer mortality, respectively [5]. Particulate matter is generally distinguished into PM-10, particles with a diameter of 10 μm or less, and PM-2.5, those 2.5 μm or less [6]. PM-10, the corresponding fine dust, primarily originates from smoke from factories or cars and is dispersed through mechanical work, while ultrafine dust corresponding to PM-2.5 primarily occurs due to chemical combinations. The agricultural sector is responsible for 17% of the total anthropogenic emission of PM10, and agricultural operations (tilling, harvesting, residue burning, etc.) have been recognized as one of the main drivers of this contribution [7]. Also, it is known that harmful fine dust to health is generated from the burning of agricultural by-products [8,9]. PM is generated through mechanical mechanisms due to farming in the soil and through chemical mechanisms of nitrogen and atmospheric substances due to the use of pesticides and fertilizers. Also, when the soil in undeveloped agricultural areas dries up without grass, fine dust is generated by turbulence due to the

wind force [10]. Airborne dust includes fine and ultrafine PMs that have become airborne due to soil erosion [11]. Fine dust blocks the stomata of vegetation, causing necrosis of cells and tissues and whitening of leaves, thereby inhibiting the growth of plants [12], and fine dust containing salt content induces salt damage resulting in withering plants [13–16].

In vast arid plains, soil aggregation, surface treatments, and windbreak facilities can be used to reduce fine dust [7,10,17–20]. Soil aggregation is a method to control the dispersion by increasing the cohesiveness of the particles, making them larger and heavier, and methods proposed include supplying moisture to the soil, coating the soil with eco-friendly bio-polymers, and stabilizing the soil using carbonates [11,20,21]. Surface treatment involves covering the soil surface with mats made of plant fibers or coating the soil surface with asphalt or concrete to prevent erosion by wind [11]. Using windbreak facilities is a method to suppress the occurrence of fine dust by reducing the wind speed on the soil surface, and to have the incoming fine dust settle to the ground. Methods include facilities like windbreak walls, windbreak fences, or using crops and trees to create windbreak forests [11,17].

To reduce fine dust in extensive open agricultural lands, using crops is the most efficient. In deciding on a fine dust reduction project, maintenance efficiency should be considered. Installing artificial facilities has effects, but it takes a lot of cost and time, and using chemical substances is not the best method considering environmental pollution. Therefore, considering initial and maintenance costs, environmental pollution, and management periods, it is advantageous to use agricultural crops in wide open field like reclaimed lands [10,22]. Reclaimed lands, especially, have a high concentration of salt in the soil, so utilizing fodder crops like barley and triticale, which can be produced even in poor environmental conditions, can contribute to soil stabilization and securing feed [23–25]. When cultivating fodder, the soil particles aggregate through the roots, and adds the crop cover, which increases the surface roughness, thus reducing the wind speed; it has the effect of reducing diffusive dust from the soil surface. When the soil moisture is high, particles tend to coalesce, which suppresses the generation of fine dust. Moreover, the surfaces of the crops in agricultural lands can adsorb fine dust, so they can act as fine dust reduction filters.

However, there are very few studies on quantifying the fine dust reduction effects by crop cultivation in agricultural lands. Some previous studies measured the difference in real-time fine dust concentrations inside and outside of forests to calculate the fine dust reduction effects of forests or directly collected fine dust adsorbed on leaves and compared the weights of the samples [26,27]. The number of fine dust particles adsorbed on the plant epidermis was counted with an electron microscope, and the fine dust reduction effect of crops was estimated based on the LAI (Leaf Area Index) [28,29]. To directly measure the amount of fine dust adsorbed on leaves, sampled leaves were washed with a brush and distilled water, and then the wash water was passed through a filter. The amount of fine dust adsorption was calculated using the weight of the filter before and after the experiment [30,31]. However, to quantitatively estimate the reduction effects of fine dust in extensive agricultural lands, it is necessary to develop coefficients for the fine dust reduction efficiency considering the characteristics of crops and to have a formula that can calculate the reduction effects during the cultivation period considering environmental conditions.

In this study, we determined the PM reduction coefficient for each crop to quantitatively present the amount of fine dust reduced by adsorption to crops in agricultural land. We also derived a method to calculate the amount of fine dust reduction during the growing period, considering the weather conditions. Focusing on barley and triticale forage crops cultivated in Saemangeum, the largest reclamation area in Korea, we analyzed the dust reduction effects of crops according to crop types and planting densities. In the field, it is challenging to prepare crops that are free of fine dust attachment, and the unstable wind conditions make it difficult to achieve desired results. Therefore, to replicate the field conditions as closely as possible, crops harvested from the field were used, and a

wind tunnel experiment was conducted using the particle size distribution of fine dust measured on-site.

## 2. Materials and Methods

To quantify the reduction effect of fine dust by crops, field monitoring and self-manufactured small-scale wind tunnel experiments were conducted (Figure 1). In the field monitoring, the vertical wind profile, fine dust particle size distribution, and crop growth were measured to determine the environmental conditions for the wind tunnel experiments. Barley and triticale were adequately sampled for the wind tunnel experiments before harvesting. To analyze the fine dust reduction effect of crops, a small wind tunnel was designed and manufactured for the purpose. To determine the fine dust reduction coefficient, two types of wind tunnel experiments were conducted: (1) the weight method, where the amount of fine dust attached to the leaves was washed and the weight was measured using a filter, and (2) the measurement of real-time concentration distribution of fine dust before and after the crop. Through field monitoring and wind tunnel experiments, the meteorological coefficient, fine dust reduction coefficient by crop type, and by density were obtained. Using these, a formula that can quantify the fine dust reduction effect of crops during the cultivation period was derived, and using this, the amount of fine dust reduction during the crop growth period in the experimental area was calculated.

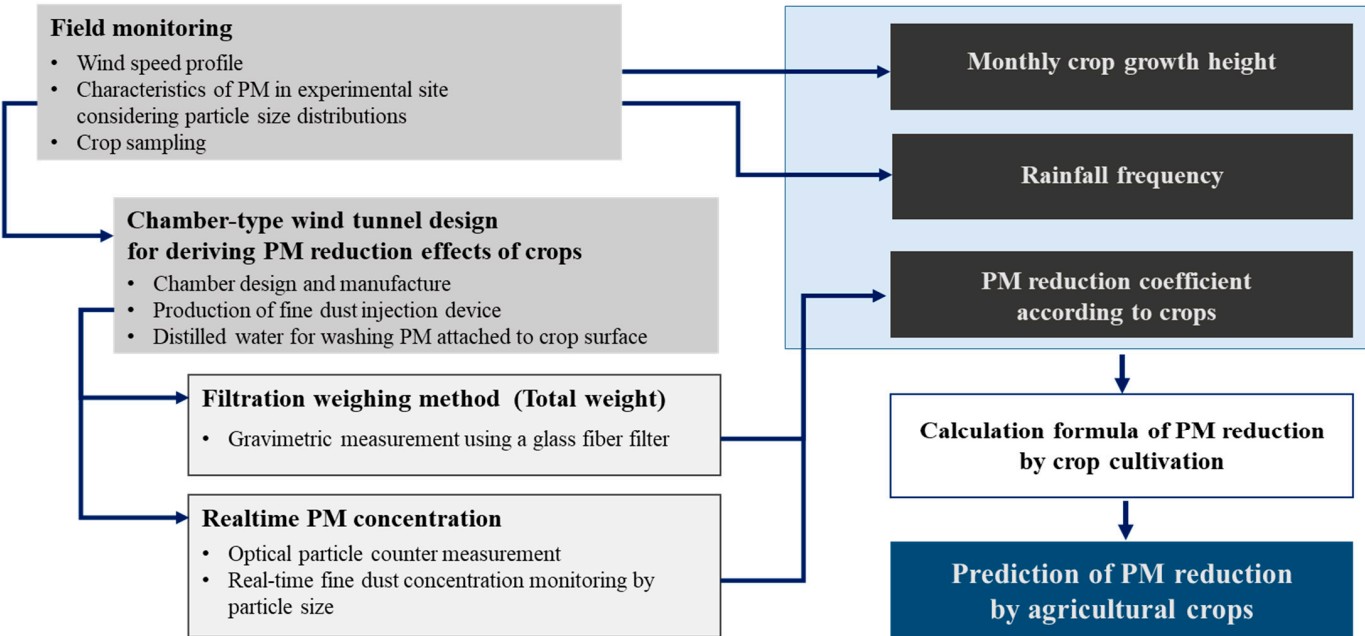

**Figure 1.** Research flow chart for calculating the PM reduction using crop cultivation in agricultural field.

### 2.1. Field Monitoring

Saemangeum in Korea is a large-scale reclaimed land with 291 km$^2$ of land and 118 km$^2$ of lakes, possessing broad and flat terrain with almost no obstacles such as mountains or hills nearby [19]. The soil of Saemangeum is saline, making plant growth challenging and resulting in low vegetation coverage. Due to its proximity to the coast and the absence of obstacles, a substantial amount of fugitive dust can originate from the soil. By cultivating halophytes, the emission of fine dust was suppressed in the initial stages of reclamation [10,13,18,32].

Field monitoring was conducted to understand the averaged wind speed and the particle size distribution characteristics of the fine dust in the agricultural life experiment complex (35.828466 N, 126.687389 E) in Saemangeum, a 20-ha site in the center of the reclaimed land. The experimental area, almost devoid of surrounding obstacles, provided

suitable environmental conditions to test the dust reduction effects due to crop cultivation. Barley and triticale were each cultivated on 2000 m², and field monitoring was performed to analyze the wind environmental characteristics of the farmland and to collect crops needed for the experiment (Figure 2). To apply wind speeds as the boundary condition in the wind tunnel experiment, they were measured at different heights from the ground surface using a multi-channel anemometer (Kamnomax anemomaster 1560, Kanomax Inc., Osaka, Japan). Upon measuring the wind speed during daytime at the experimental reclaimed land, the average wind speeds at heights of 25, 50, 100, and 150 cm from the ground were found to be 1.87, 2.45, 3.10, and 3.75 m/s, respectively, with a maximum wind speed of 5.5 m/s being observed at 150 cm height. Considering the height of the crops, the wind tunnel experiments were conducted using the 3.10 m/s wind speed measured at 100 cm as the representative wind speed. During the experimental period, the on-site weather conditions were measured by installing an automatic weather station (AWS, U30, Onset Inc., Bourne, MA, USA), and the annual weather information was obtained using data from the nearest official meteorological station. For the wind tunnel experiment, barley and triticale crops that had grown more than 1 m were sampled.

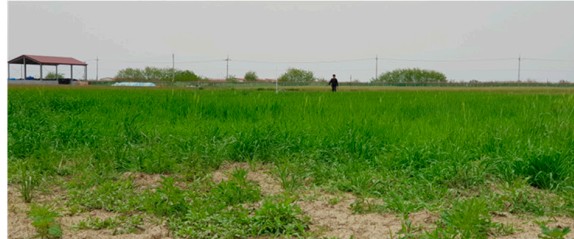

(a) Barley cultivation area

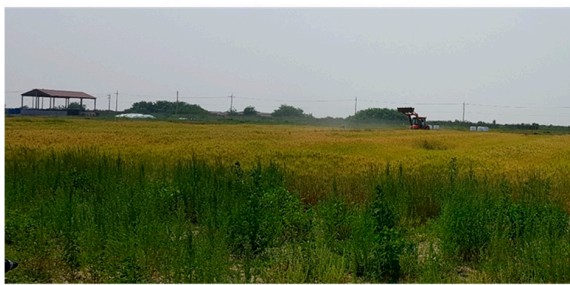

(b) Triticale cultivation area

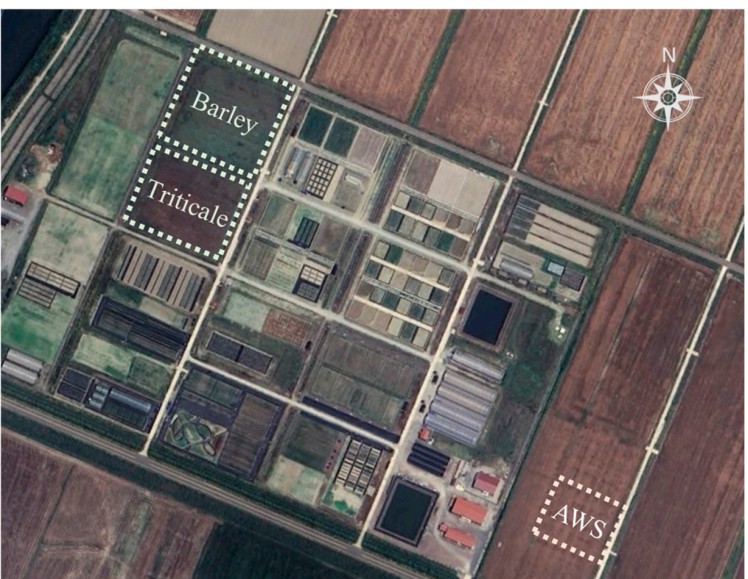

(c) Satellite image for experimental site

**Figure 2.** Experimental site for field monitoring with cultivation of (**a**) barley and (**b**) triticale in the (**c**) crop cultivation test areas located in the Saemangeum reclaimed land in Korea.

### 2.2. Experimental Wind Tunnel Design

To analyze the ability of crops to reduce fine dust, a small wind tunnel was designed and constructed. In field experiments, environmental conditions continuously change, making it very challenging to conduct stable experiments to determine dust reduction coefficients. To analyze the fine dust reduction effect of crops, the concentration of fine dust must be measured before and after it passes through the crops. Since the wind tunnel has a constant and stable wind speed, and the space where the air moves is confined, it is possible to stably measure the concentration of fine dust before and after passing through the crops.

The wind tunnel was manufactured with a height of 1 m, width of 0.5 m, and length of 10 m, and two variable fans with a maximum air volume of 4740 m³/h were installed to enable negative pressure ventilation (Figure 3). To prevent fine dust from sticking to the wind tunnel surface, the interior walls made of particle board were coated with Teflon film. To facilitate the installation of crops for the experiment and cleaning before and after the experiment, each component of the wind tunnel was designed as a detachable modular type with wheels attached. The fine dust injection device was installed by combining a

pump so that air mixed with fine dust in a Tedlar bag could be introduced and mixed into the wind tunnel at a constant flow rate. The temperature, humidity, and flow rate inside the wind tunnel were analyzed using an indoor environment measuring instrument (AMI 310, KIMO, Saint-Priest, France).

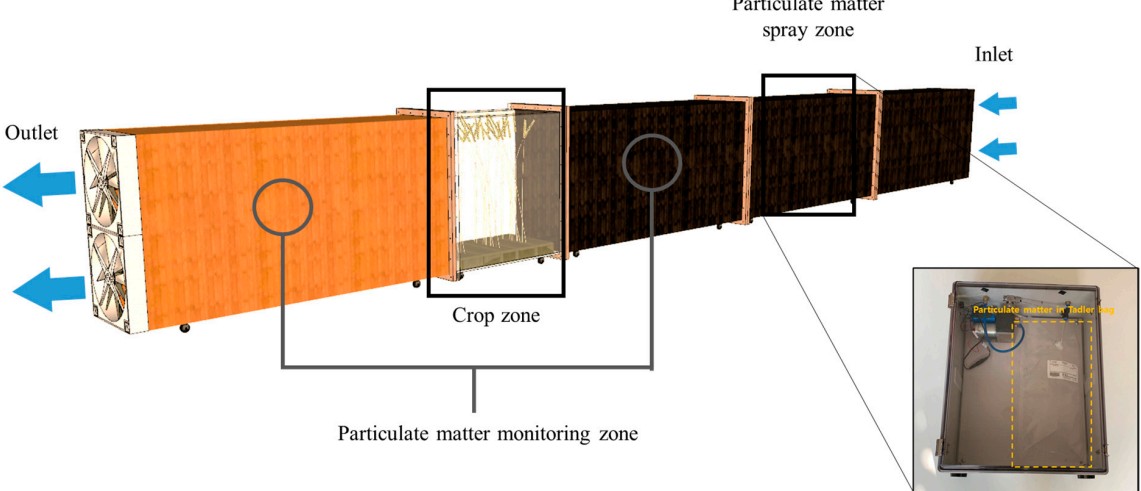

**Figure 3.** Design of a small wind tunnel to analyze the fine dust reduction coefficiency according to crops and its densities.

Barley and triticale were sampled just before harvest in the experimental area and used for the wind tunnel test. The harvested crops were carefully washed using triple distilled water to clean the fine dust adsorbed on the crop surface without damaging the crops; and after washing, they were naturally dried in preparation for the experiment. The prepared crops were installed in the crop zone of the wind tunnel, and the arrangement of the crops was performed at 5 cm intervals according to the sowing standards provided by the Rural Development Administration in Korea. The density of sowing was categorized as densely, moderately, and sparsely sowing, each placed at intervals of 20, 25, and 35 cm, respectively (Figure 4).

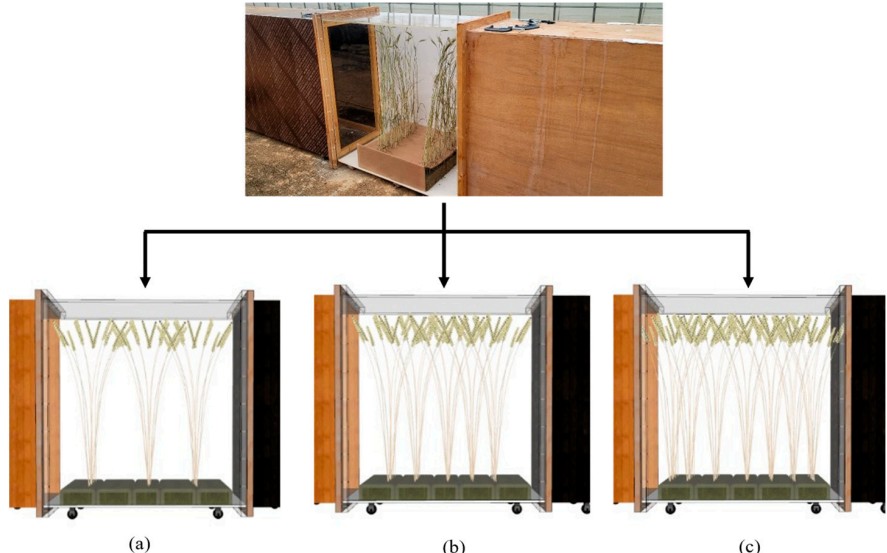

**Figure 4.** Arrangement according to crop density for calculating dust reduction coefficient through the wind tunnel experiment. (**a**) Densely crop sowing, (**b**) Moderately crop sowing, (**c**) Sparsely crop sowing.

The size distribution of particulate matter at the reclaimed land site was monitored by using OPCs (Optical Particle Counter, 11-D, Grimm Inc., Freising, Germany), which can measure aerosol concentrations divided into 31 sizes ranging from 0.253 μm to 35.15 μm at 6-s intervals. Capturing the fine dust from the site and reproducing it in the experimental chamber was very challenging. Therefore, using the measured size distribution, the experimental fine dust was combined by two standard dusts to achieve a particle size distribution as similar to the results measured on-site (Figure 5). The standard dusts used were nominal 0–3 micron Arizona test dust (Powder Technology Inc., Arden Hills, MN, USA) and A1 test dust (ISO 12103-1 [33], Powder Technology Inc., Arden Hills, MN, USA). The size distribution of fine dust measured in the field showed the highest proportion at 5.5~11 μm range, and mixing the two types of standard fine dusts at various ratios showed the most similar size distribution at a 5:5 ratio (Figure 5).

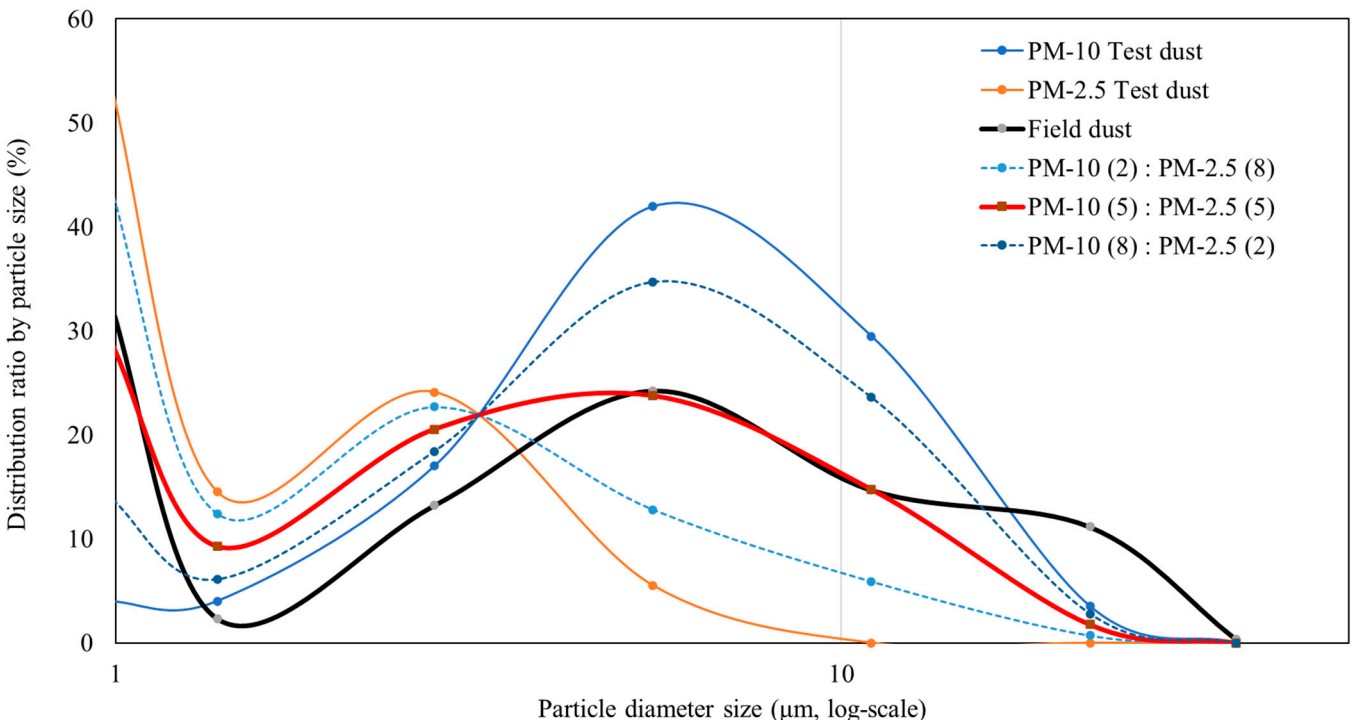

**Figure 5.** Size distribution of fine dust concentration by measured and manufactured fine dust for the wind tunnel experiment: composition ratio of test dust by particle size according to the composition ratio of standard dust of PM-10 and PM-2.5.

*2.3. Determination of PM Reduction Coefficient*

To calculate the PM reduction coefficient of the crops, the gravimetric method and real-time concentration analysis method were carried out. The first experiment involved directly measuring the amount of PM adsorbed on the surface of the crops after providing fine dust as similar as possible to field environmental conditions. For the experiment, barley and triticale surfaces were carefully washed with distilled water and completely dried, then fixed and placed in the test section considering three different planting densities (Figure 6). The manufactured particulate matter was mixed with air and supplied steadily at a flow rate of 1 L/min through a spraying device, with 1 mg/L of fine dust. After sufficient time of the particulate matter supply had concluded, the crops were washed using triple-distilled water to clean the adsorbed particulate matter on the surface. Impurities such as plant hairs or epidermal cells included in the washing water were filtered out using a filter with 100 μm pore size. Subsequently, a filtering apparatus was fabricated by combining filtering apparatus (LT. SS-47S, Korea Ace Scientific, Ltd., Seoul, Republic of Korea) and a 47 mm glass fiber filter (1.0 μm pore size, PALL 61631, Pall Corp., Washington, DC, USA), and

the particulate matter included in the washing water was separated through the filter and pump. The filtered particulate matter was then dried for 24 h in a desiccator, and the weights of the filters before and after the experiment were accurately measured using Micro Analytical Balances (BM-22, AND Weighing Inc., Tokyo, Japan). All experiments were conducted three times under independent conditions, and blank sample filters that did not capture particulate matter were used for correction.

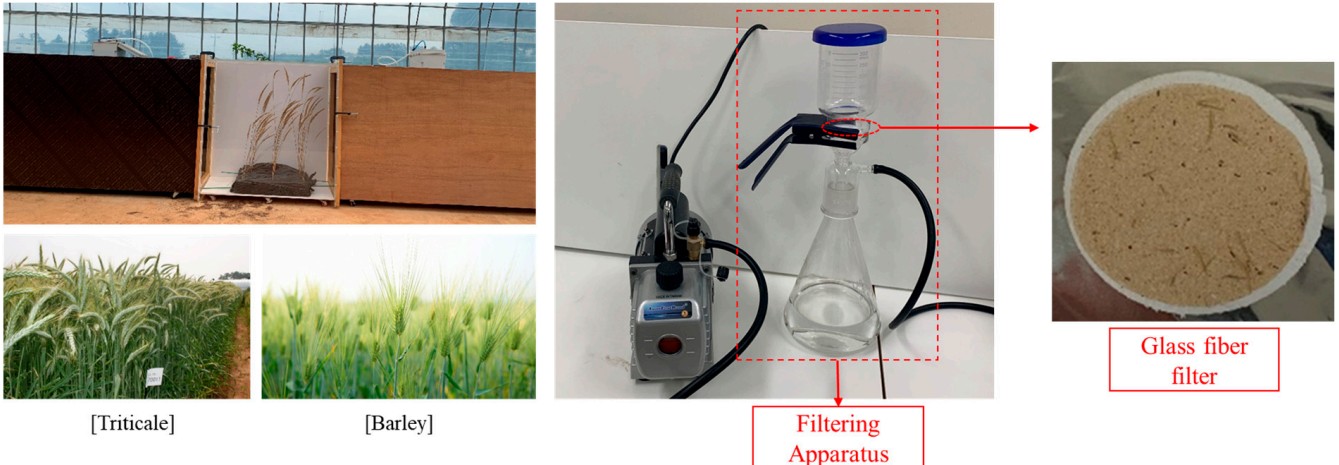

[Triticale]　　　　　　　[Barley]　　　　　　　Filtering Apparatus　　　　Glass fiber filter

**Figure 6.** Filtration system for wash water and filtered samples for determining particulate matter reduction coefficient.

For the second experiment to determine the PM reduction coefficient of crops, real-time and size-specific monitoring of dust concentrations were performed before and after passing through crop canopy. When using filters by gravimetric methods, it is possible to measure the amount of particulate matter attached to the crop surface, but it is challenging to analyze the size distribution and the particle characteristics. Therefore, optical particle counters (OPC) were installed at three locations within the chamber: at the inlet, and before and after the crop zone. Monitoring was carried out at 6-s intervals for real-time and 31 different sizes of particulate matter. Figure 7 shows the results of continuous monitoring of PM-10 and PM-2.5 before and after the crop zone for moderately sown triticale for 100 min. The real-time concentrations of particulate matter were measured by size, and the data were corrected using the concentration of particulate matter entering from the outside air. It was assumed that the amount reduced or adsorbed by the crops corresponds to the difference in concentration before and after passing through the crops. The results for the two crops and three planting densities were refined using the same procedure, and the difference in particulate matter concentrations before and after the crop zone was used to calculate the reduction amount over time for PM-10 and PM-2.5 separately. The cumulative reduction amount over time was used to derive the PM reduction coefficient.

*2.4. Calculation of PM Reduction Amount*

The PM reduction coefficient indicates how much particulate matter can be adsorbed and reduced by crops with clean surfaces. However, the adherence of particulate matter to crop surfaces is not a continuous occurrence. Once a sufficient amount of particulate matter adheres to the entire crop surface, there will be no additional reduction effect, as the amount of particulate matter redispersed by the wind and the amount adhering to the leaves balance each other. After this, when rain washes away the particulate matter again, the particulate matter starts to adhere at the crop surface again, repeating the cycle from the beginning. Therefore, to analyze the amount of particulate matter reduced during the cultivation period, the rainfall frequency, which can consider the influence of rainfall during the entire period, was calculated. The particulate matter reduction effect of crops

was determined using the crop canopy over the cultivation period, rainfall frequency, and particulate matter reduction coefficient.

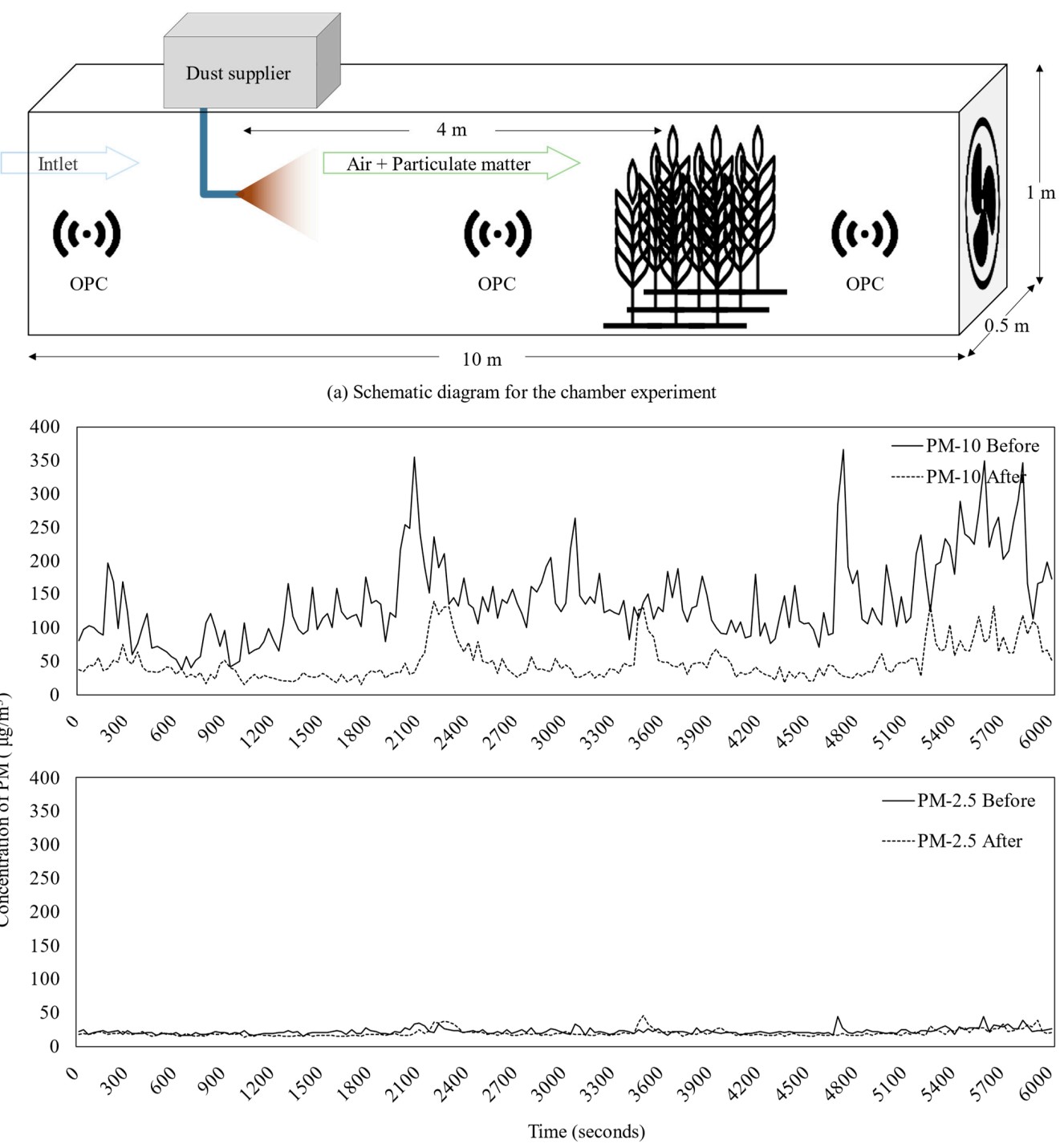

(a) Schematic diagram for the chamber experiment

(b) PM concentrations before and after crop canopy

**Figure 7.** (**a**) Schematic diagram of wind tunnel test for deriving PM reduction coefficient and (**b**) PM-10 and PM-2.5 concentration changes before and after passing through crop canopy moderately sown triticale for 100 min.

## 3. Results

### 3.1. PM Reduction Coefficient by Wind Tunnel Experiments

To determine the PM reduction coefficient due to crop absorption, according to cultivation density for triticale and barley, a wind tunnel experiment was performed. After injecting fine dust for a sufficient time, the particulate matter adhered at the crop surface was washed with triple-distilled water, then filtered to precisely measure its weight, thereby determining the adsorption amount. Figure 8 represents the results, converted into the amount of particulate matter absorbed per hectare for triticale and barley. Based on moderately sowing, the particulate matter absorption amount of the crop showed that triticale was about 1.21 times higher than barley, and based on densely sowing, it was 1.34 times higher. Triticale could reduce more particulate matter as it has more crop hairs on its surface, and thus, a relatively larger adsorption surface area. Analysis results according to planting density showed that in the case of sparsely sowing, triticale decreased by 50.0% and barley by 79.8% in particulate matter absorption amount. This is because when crops are sparsely planted, the number of crops that can adsorb particulate matter decreases, space occurs between the crops, and the shaking range of the crops increases, causing the amount of resuspended particulate matter to increase, thus sharply lowering the reduction efficiency. However, triticale showed a 3.02 times higher reduction effect compared to barley, even in sparsely sowing, because the hairs on the surface reduced the amount of resuspension. The PM reduction coefficient of the crops by the gravimetric method was derived as 2233 g/ha for triticale and 1839 g/ha for barley, based on moderately sowing. However, it is calculated based on the crop cultivation area, so it cannot take into account the changes in canopy due to the growth of the crops.

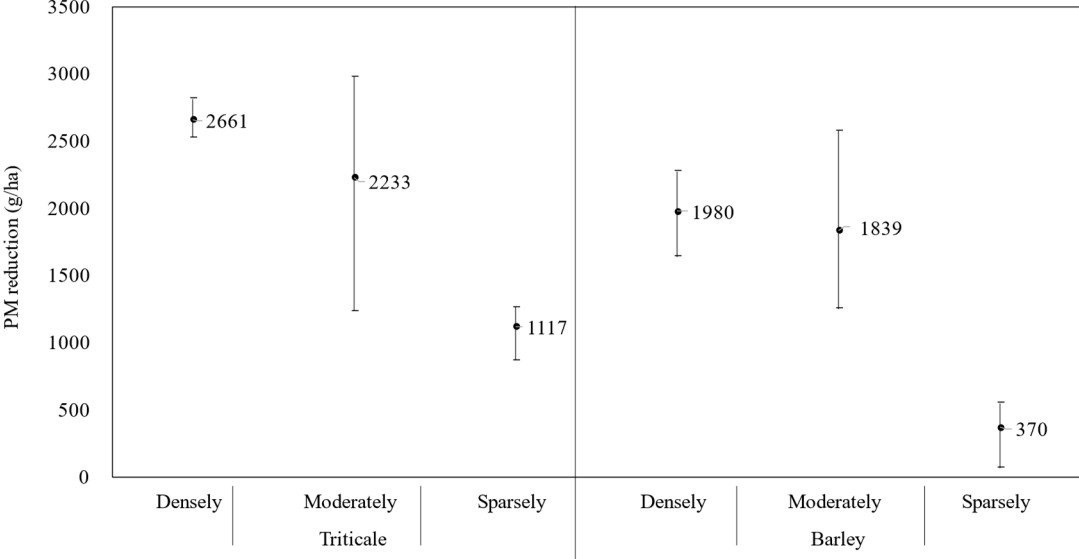

**Figure 8.** The PM reduction coefficient based on measuring the fine dust attached to the surface of crops using the gravimetric method according to two crops (triticale and barley) and its cultivation density.

The second method to determine the PM reduction coefficient is to analyze the real-time concentration of airborne fine dust before and after passing through the crop zone. The incoming air from the outside to wind tunnel was used to calibrate the data, and through the wind tunnel experiments on the planting density of triticale and barley, the difference in PM-10 and PM-2.5 concentrations before and after the crop zone was presented over time (Figure 9). Because the experiment started with crops that were completely washed and dried, a large amount of fine dust was absorbed initially after the start of the experiment, showing a high PM reduction amount. As time passed, the reduction amount of fine dust

showed a decreasing trend in a logarithmic graph. After a certain time, a section where the reduction amount of fine dust of crops became constant appeared. When the crops were densely installed, it took more time for the reduction concentration to reach 0 due to more fine dust being able to be absorbed at the crop surface. The reduction amount over time at the bottom of the graph was integrated to derive the PM reduction coefficient for each crop and planting density (Table 1).

**Table 1.** PM reduction coefficients for PM-2.5 and PM-10 by crop types and planting densities, converted based on real-time concentration differences before and after crop canopy in the chamber experiments (unit: mg/m$^3$).

| Crop | Density | PM-2.5 | PM-10 |
|------|---------|--------|-------|
| Triticale | Sparsely | 7.0 | 34.0 |
| | Moderately | 120.0 | 82.2 |
| | Densely | 254.7 | 400.3 |
| Barley | Sparsely | 3.1 | 9.5 |
| | Moderately | 14.5 | 26.9 |
| | Densely | 15.2 | 92.7 |

The PM reduction coefficient for each crop was calculated based on the amount of fine dust absorbed by the crop canopy, and this can be used to calculate the amount of PM reduction in the entire agricultural land. The PM reduction coefficient for triticale, depending on whether the planting density was sparse, moderate, or dense, was 7.0, 120, and 254.7 mg/m$^3$ for PM-2.5, and 34.0, 82.2, and 400.3 mg/m$^3$ for PM-10, respectively. In the case of barley, it was 3.1, 14.5, and 15.2 mg/m$^3$ for PM-2.5, and 9.5, 26.9, and 92.7 mg/m$^3$ for PM-10, respectively. At general sowing intervals, triticale showed a higher PM reduction coefficient than barley, with PM-2.5 being 8.3 times higher and PM-10 being 3.1 times higher. This is because triticale, being a relatively hairy crop, could have fine dust attached to more surfaces, and especially when triticale is densely planted, the intertwined hairs of the crops can act like a high-efficiency filter, and accordingly, the PM reduction effect for PM-10 rose up to 4.87 times compared to the general planting density. This was also the case for barley, where the fine dust reduction effect increased by 3.45 times when densely planted.

### 3.2. Estimation of PM Reduction Effects by Crops

To calculate the cumulative PM reduction according to the cultivation of experimental crops, we utilized PM reduction coefficients, canopy variations due to crop growth, and rainfall frequency. The PM reduction coefficient was determined through wind tunnel experiments targeting fully grown crops, but the canopy changes during the cultivation period due to crop growth and planting density. To reflect the crop density and the height, growth curves were derived based on the previous studies on the height of crops such as triticale and barley and the measured height of crops in the field [34]. The growth stages used average values by month, and the crops, sown in October, grew about 1 m height by the time they were harvested the next June (Figure 10). Rainfall frequency was analyzed from meteorological data for the Saemangeum reclaimed land, the experimental area, from October 2021 to June 2022. Fine dust can reduce the amount in the air by adhering to the crop surface and then being washed away by rainfall, repeating this process until the next rainfall. After sufficient rain washes away the fine dust on the crop surface, it can have a reduction effect again, like a washed filter; hence, rainfall frequency was calculated through weather analysis. The frequency of rainfall was calculated based on the monthly frequency of days when more than 0.5 mm of rain per hour was recorded, reflecting the measurement level of rain gauges. It was counted as one occurrence when it rained continuously for several hours or days.

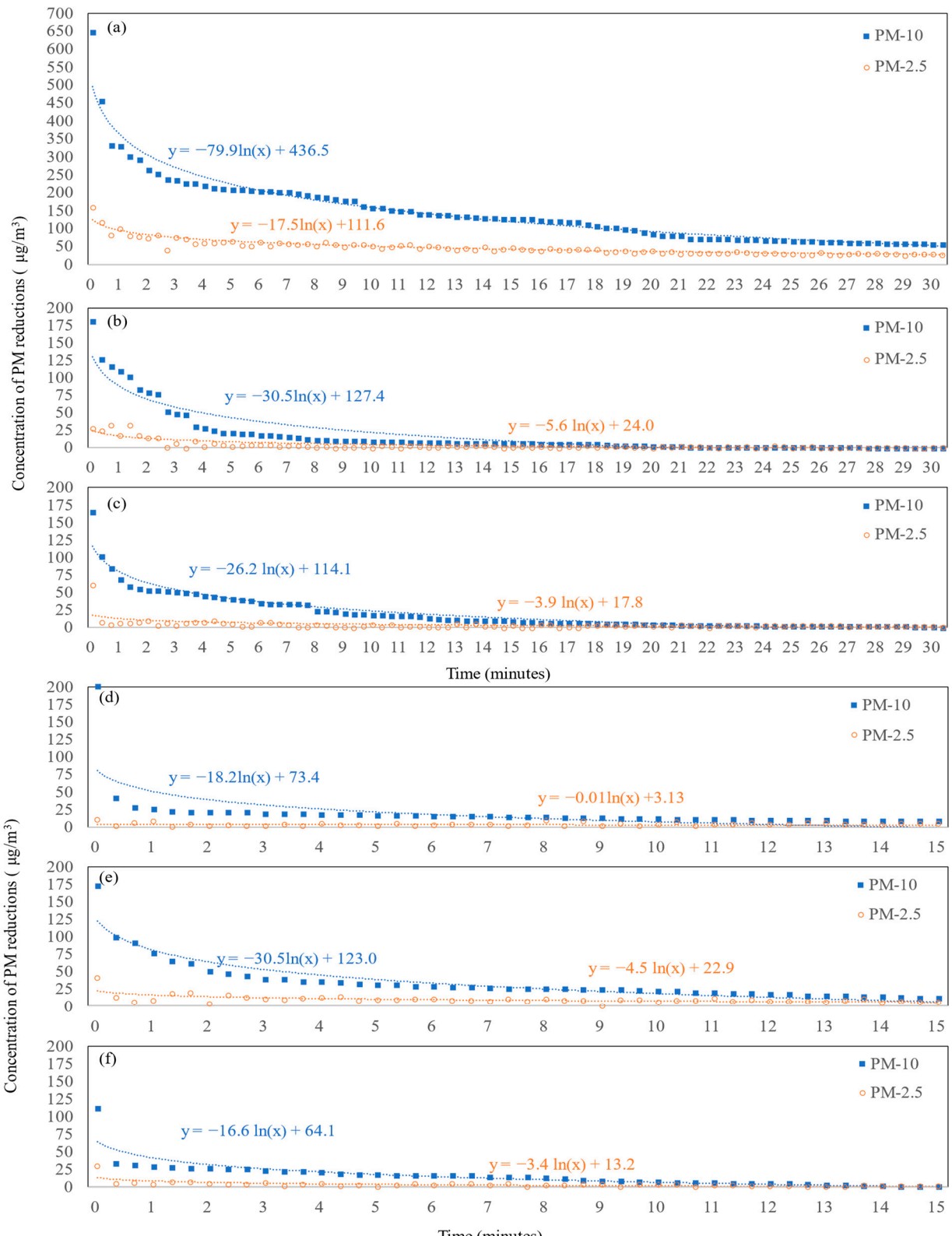

**Figure 9.** The PM reduction effect of triticale over time, determined based on the difference in PM concentration before and after passing through the crop zone with difference crop sowing densities. (**a**) Densely triticale sowing, (**b**) Moderately triticale sowing, (**c**) Sparsely triticale sowing, (**d**) Densely barley sowing, (**e**) Moderately barley sowing, (**f**) Sparsely barley sowing.

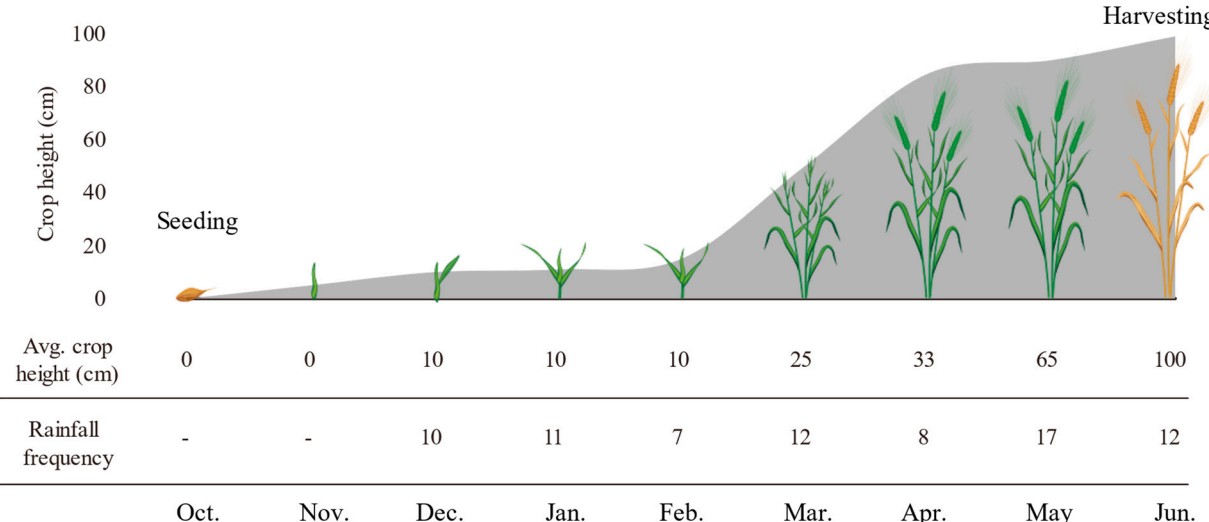

**Figure 10.** Average growth curves of Triticale and Barley measured in the field and rainfall frequency determined through meteorological data.

To calculate the cumulative overall PM reduction with triticale and barley during the cultivation period, the cumulative reduction of fine dust was calculated based on the crop canopy, rainfall frequency, and PM reduction coefficient for eight months after the leaves came out (Figure 10). The PM reduction was calculated considering the crop canopy and weather conditions during the cultivation period in the experimental reclaimed land. The PM reduction in agricultural fields according to the crop types and densities was determined monthly using Equation (1) and accumulated over the cultivation period to calculate the total reduction effect. Based on the monthly average height of the crop ($h_c$, m) and the area of the agricultural field ($A$, m$^2$), the crop canopy (m$^3$) was determined. This was then multiplied by the reduction coefficient for each crop ($R_{crop}$, mg/m$^3$·time) in Table 1 and the frequency of rainfall ($F_{rain}$, time/month) to make the monthly PM reduction amount.

$$PM\ reduction = \Sigma\left(h_c \times A \times R_{crop} \times F_{rain}\right) \tag{1}$$

The monthly PM reduction was accumulated to calculate the PM reduction during the crop cultivation period (Figure 11). Comparing triticale and barley, which are cultivated during the same period, triticale, which has relatively more hairs, showed a greater reduction in PM. In the typical cultivation density used in the experimental reclaimed area, triticale reduced 126.1 kg/ha of PM-10 and 80.2 kg/ha of PM-2.5. In contrast, barley, cultivated during the same period, showed a reduction effect of 29.2 kg/ha for PM-10 and 4.8 kg/ha for PM-2.5.

If the crop density is low, the space between the crops becomes wider, and the extent to which the crops are shaken by the wind increases, resuspending the attached fine dust. If the crops are sown sparsely, the adhesion amount decreased by 71.7% and 64.7% for triticale and barley, respectively, based on PM-10. Conversely, when the crops are sown densely, the crops could act as a filter with less shaking, absorbing more fine dust, showing a result of 3.34 times and 3.44 times increases in the amount of PM reduction in triticale and barley, respectively, based on PM-10.

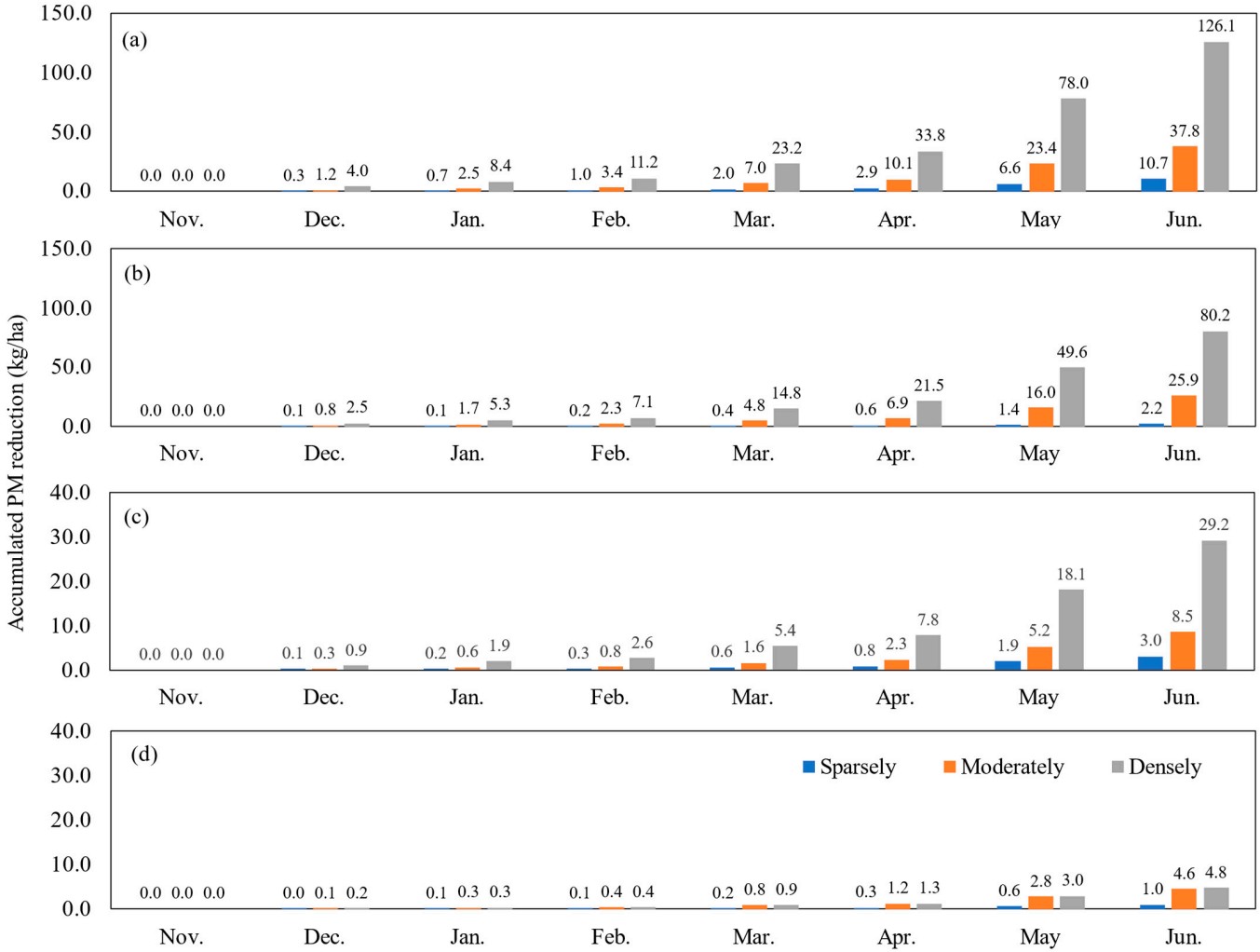

**Figure 11.** Accumulation of fine dust reduction per hectare according to crop type, monthly canopy and rainfall frequency. (**a**) Triticale (PM-10), (**b**) triticale (PM-2.5), (**c**), barley (PM-10) (**d**) barley (PM-2.5).

## 4. Discussion

Fine dust can have serious effects on health, causing respiratory diseases, weakening the immune system, and affecting crop growth. To reduce fine dust over large developed or agricultural areas, it is more efficient to use crops than machinery or equipment. Fodder crops, which can grow even in barren and rough environments, can reduce airborne dust by coagulating soil particles and reducing wind speed on the surface and have the effect of reducing fine dust by adsorbing it on the crop surface. In this study, the PM reduction coefficient of crops was derived through a chamber-type wind tunnel experiment to quantitatively assess the dust reduction effect of crops by type and planting density. Additionally, a PM reduction formula considering crop growth and weather conditions during the cultivation period was derived.

A small wind tunnel was designed and manufactured to establish stable environmental conditions following the environmental conditions obtained through field experiments. The PM reduction coefficient was measured by two experiments: the gravimetric method, which weighs the dust adhered to the leaves washed with distilled water using a filter, and measuring the real-time diameter distribution of dust concentration before and after the crop. Based on the experimental results, the effect of reducing fine dust during the cultivation period was calculated, considering the dust reduction coefficient, rain frequency, and the canopy of the crops.

During the cultivation period, the PM reduction amount was calculated using the rainfall frequency, crop canopy, and PM reduction coefficient. According to the gravimetric method, the dust reduction coefficient of the crops, with moderately sowing as a standard, showed triticale at 2233 g/ha and barley at 1839 g/ha. The crop-specific PM reduction coefficient calculated through real-time concentration analysis showed triticale at PM-2.5 82.2 mg/m$^3$, PM-10 120 mg/m$^3$, and barley at PM-2.5 14.5 mg/m$^3$, PM-10 26.9 mg/m$^3$ under general planting density. Triticale, having relatively more hairs than barley, showed higher PM reduction coefficients, 8.3 times for PM-2.5 and 3.1 times for PM-10 at general sowing intervals. For the experimental target area, PM reduction coefficients, the canopy varying with crop growth, and rain frequency were used to calculate the cumulative dust reduction amount due to crop cultivation. During the general planting density cultivation period of triticale and barley, PM-10 was reduced by 37.8 kg/ha and 8.5 kg/ha, respectively, and PM-2.5 was reduced by 25.9 kg/ha and 4.6 kg/ha, respectively. The dust reduction effect during the cultivation period was up to 126.1 kg/ha in terms of PM-10 when triticale was cultivated with densely sowing planting density.

### 5. Conclusions

This study has proposed a new method that can quantitatively calculate the amount of reducing particulate matter during the cultivation period of fodder crops. Through wind tunnel experiments, the amount of fine dust adhering to the surface of crops was measured using the gravimetric method, similarly used in previous studies. At the same time, real-time particle size distributions were measured by optical particle counters according to crops and sowing density, which were lacking in existing research. The PM reduction coefficient was derived from the concentrations before and after passing through the crop group. A new analytical method has been proposed that can derive reduction effects considering rainfall frequency in the field and growth of crops. In the future study, this can be expanded to analyzes of various types of crops, and can be used to propose cultivation densities and methods that can increase the efficiency of reducing fine dust by utilizing simulation models such as computational fluid dynamics.

**Author Contributions:** Investigation, K.A.K.; Writing—original draft, S.-W.L.; writing—review and editing, I.-H.S. All authors have read and agreed to the published version of the manuscript.

**Funding:** This work was carried out with the support of "Cooperative Research Program for Agriculture Science & Technology Development (Project No. PJ0170752022023)" Rural Development Administration, Republic of Korea.

**Informed Consent Statement:** Not applicable.

**Data Availability Statement:** Data is available from the corresponding author on request.

**Conflicts of Interest:** The authors declare no conflict of interest.

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
