# Peer review of "Fine Particulate Matter Reduction through Crop Surface Adsorption in an Agricultural Field Using the Coefficient Derived from Wind Tunnel Experiments"

_applsci, doi:10.3390/app132112072_

Round 1
Reviewer 1 Report
Comments and Suggestions for Authors
Dear Authors,
At the outset, I congratulate for the conceiving this idea
Few questions needs to addressed before it is published
1) How does moisture in the soil and Ambient RH content effects the PM resuspension in the crop lands
2) What are the particle size bins measured in this experiments
3) How does the OPC analysers are calibrated? if possible please share the details
4) How can the comaprision been drawn between the wind tunnel experiment conditions and the real time field measurements.
5) Does the experiments includes the pollen suspensions of the crops
Comments on the Quality of English Language
NA
Author Response
We deeply appreciate your precious reviewed in this manuscript, thus we can improve this research article. We revised and corrected refer to your comments.

Reviewer 2 Report
Comments and Suggestions for Authors
Fine dust is one of the most important factors of air quality which may harm human health. Fortunately, crops can avoid this threat by coagulating soil particles and reducing wind speed on the surface to a certain extent. Based on concepts dust reduction coefficient of crops, the authors develop a self-manufactured wind tunnel experiment to quantitatively assess the dust reduction and analyze the effect of type and planting density. This work is interesting and useful for dust control. However, I think the authors need to make some major modifications in analysis, data and other aspects to meet the basic requirements of this journal.
Major comments:
1. Was this wind tunnel built and used for the first time in this work? If so, internal wind field quality information is required to assess whether the wind tunnel is up to standard. In addition, it is also necessary to give a comparison between wind field data and wind tunnel data, as mentioned by the author in lines 138-139. I want to know if the wind speed, temperature and other data in the wind tunnel are consistent with the field.
2. I don't think the article is complete and I haven't seen what the author claimed “a dust reduction formula considering crop growth and weather conditions during the cultivation period”. Qualitative description is far from sufficient in lines 355-362.
3. Authors should compare the two measurement methods in lines 133-135 to verify accuracy and compare advantages and disadvantages.
Minor comments:
1. It is necessary to give a detailed description of the optical particle counters (OPC) instrument, or to give a reference link at least.
2. In Figure 7b, it is necessary to clarify what kind of working condition is it? In addition, which two probes are used for the curve in the figure?
3. The authors claim that they are able to measure particle size distribution in real time in line 415. With what instruments? What's the principle?
Author Response

(The authors gave the same response as above.)

Round 2
Reviewer 1 Report
Comments and Suggestions for Authors
I can be accepted
Comments on the Quality of English LanguageI can be accepted
Reviewer 2 Report
Comments and Suggestions for Authors
The authors have answered all my questions.